# Routing Density Analysis of Area-Efficient Ring Oscillator Physically Unclonable Functions

**Zulfikar Zulfikar** [1,2,*], **Norhayati Soin** [2,3], **Sharifah Fatmadiana Wan Muhamad Hatta** [2,3], **Mohamad Sofian Abu Talip** [2] and **Anuar Jaafar** [4]

1 Department of Electrical and Computer Engineering, Faculty of Engineering, Universitas Syiah Kuala, Jl. Teuku Nyak Arief, Darussalam, Banda Aceh 23111, Indonesia

2 Department of Electrical Engineering, Faculty of Engineering, Universiti Malaya, Kuala Lumpur 50603, Malaysia; norhayatisoin@um.edu.my (N.S.); sh_fatmadiana@um.edu.my (S.F.W.M.H.); sofian_abutalip@um.edu.my (M.S.A.T.)

3 Center of Printable Electronics, Universiti Malaya, Kuala Lumpur 50603, Malaysia

4 Centre for Telecommunication Research & Innovation (CeTRI), Fakulti Kejuruteraan Elektronik dan Kejuruteraan Komputer (FKEKK), Universiti Teknikal Malaysia Melaka (UTeM), Melaka 76100, Malaysia; anuarjaafar@utem.edu.my

* Correspondence: zulfikarsafrina@unsyiah.ac.id

**Abstract:** The research into ring oscillator physically unclonable functions (RO-PUF) continues to expand due to its simple structure, ease of generating responses, and its promises of primitive security. However, a substantial study has yet to be carried out in developing designs of the FPGA-based RO-PUF, which effectively balances performance and area efficiency. This work proposes a modified RO-PUF where the ring oscillators are connected directly to the counters. The proposed RO-PUF requires fewer RO than the conventional structure since this work utilizes the direct pulse count method. This work aims to seek the ideal routing density of ROs to improve uniqueness. For this purpose, five logic arrangements of a wide range of routing densities of ROs were tested. Upon implementation onto the FPGA chip, the routing density of ROs are varied significantly in terms of wire utilization (higher than 25%) and routing hotspots (higher than 80%). The best uniqueness attained was 52.71%, while the highest reliability was 99.51%. This study improves the uniqueness by 2% subsequent to the application of scenarios to consider ROs with a narrow range of routing density. The best range of wire utilization and routing hotspots of individual RO in this work is 3–5% and 20–50%, respectively. The performance metrics (uniqueness and reliability) of the proposed RO-PUF are much better than existing works using a similar FPGA platform (Altera), and it is as good as the recent RO-PUFs realized on Xilinx. Additionally, this work estimates the minimum runtimes to reduce error and response bit-flip of RO-PUF.

**Keywords:** area-efficient ring oscillator; routing density; routing hotspots; runtimes



## 1. Introduction

Field Programmable Gate Arrays (FPGAs) integrated circuits (ICs) are a crucial key electronic component embedded in today's sophisticated electronic systems, primarily due to their excellent programmable and reconfigurable capabilities combined with relatively low non-recurring engineering cost (NRE) and short design cycle. FPGAs' advantages quickly supersede that of application-specific of IC (ASIC) [1,2]. Additionally, the inclusion of specialized hardware blocks, such as hardcore, and the ability to implement softcore in the FPGA makes the FPGA more commonly adopted in many critical application domains, such as industrial, spacecraft, automotive, networking, and prototyping. However, as with many other semiconductor devices and integrated circuits, FPGAs also suffer from induced errors and reliability issues, which may jeopardize confidential and sensitive information. Therefore, it is paramount to guarantee the electronic system's security by ensuring the

FPGA device is reliable against the variation of operating parameters, namely temperature and static signal probability.

Physically unclonable function (PUF) is a fundamentally new hardware security system that promises a model shift in numerous security applications; its relatively simple architecture can avoid various data breaches. Some security mechanisms have currently used the concept of PUF on deep-submicron silicon materials due to the presence of intrinsically unique [3]. The main principle of PUF is to take advantage of its self-existing variations in integrated circuit production processes. The difference, such as the delay of each transistor, is random and unpredictable. Therefore, there are no identical chips in nanoscale technology. Due to this, PUF is suitable to be the digital identity of an integrated circuit or device [3].

Various types of PUF have been developed, though detailed analyses of stability and reliability are substantially lacking. Among others, PUF can be classified as follows: memory-based PUF; PUF based on delay [4]; and physical PUF. However, the grouping of the PUFs is not finalized, as researchers are continuously looking for new and better PUF models. The following are some of the PUF models in which operations are based on the principle of memory; SRAM-PUF [5], latch PUF [6], butterfly PUF [7], and flip flop PUF [8–10]. Meanwhile, PUF, the operation of which is based on delay, is the ring oscillator PUF (RO-PUF) [11,12] and arbiter PUF [13]. The RO-PUF improvement research continues due to its simple structure, ease of generating responses, and its promises of primitive security.

## 2. Existing Works

The fundamental principle of PUF based on ring oscillators has been know since 2002 [14], and since then, it has been implemented by differentiating each device's specific characteristics based on its unique frequency. Since then, many researchers have developed the concept of RO-PUF. Rahman et al. proposed the first aging-resistant RO-PUF design, which they referred to as ARO-PUF. Over time, the PUF proved to produce more reliable output [15]. Next, an improvement of RO-PUF's uniqueness and reliability parameters by manipulating the structure of ROs was developed. The authors provided an algorithm that can dismiss the systematic variation effect on RO-PUF and, in turn, enhances the security and reliability of the RO-PUF [12]. The low power RO-PUF using dynamic style feedthrough logic (FTL) to enhance reliability was carried out in 90 nm CMOS technology in 2020 [16].

Other research projects into RO-PUF include a novel current-controlled CRO PUF (CCRO-PUF) in which inverters of RO use different logic styles for security improvement. The idea is to use static CMOS and FTL. Simulations were carried out in the Cadence Virtuoso environment using SPECTRE SPICE using a library from UMC foundry of CMOS 90 nm process [17]. Subsequently, a crossover RO PUF to improve flexibility and reliability and reduce hardware overheads was also published. The basic idea was to implement one-to-one input–output mapping with a lookup table (LUT)-based interstage crossing structure in each level of inverters [18]. Barbareschi et al. proposed the frequency signature-based PUF (FS-PUF), an essential mechanism for authentication and identification of digital devices based on ROs immunized to working conditions and aging [19].

However, a substantial study has yet to be carried out in developing designs of the FPGA-based RO-PUF, which effectively balances performance and area efficiency. Firstly, most proposed RO-PUF requires a massive area where the number of RO placed on a chip is considerable. An additional area is required to realize the challenge–response pairs' (CRP) generation and other controlling circuits [20–22]. Secondly, there exist unstable ROs in the FPGA [14,15,17,18], leading to degraded reliability. Thirdly, the routes created by the Electronic Design Automation (EDA) might cause an imbalance in routing density in some locations. Ikeda et al. discussed the routing issue that affects the RO performance, specifically reducing uniqueness [23]. The issue was later discussed further by Giechaskiel

et al. by showing that the RO signal frequency is reduced due to interference by other signals [24,25].

Therefore, it is required that an RO-PUF be realized with a lesser area, which will result in good randomness and resistance upon environmental variation. This work proposes a modified RO-PUF design that connects ROs to counters directly. The proposed design can generate more response bits than the previous designs while using a smaller on-chip area. Typically, in most RO-PUF designs, the actual frequency of each RO differs because of physical variation. However, in nanoscale technology, diversity is becoming less and challenging to detect. Moreover, the routing delay is dominant compared to the logic delay itself. Therefore, upon realizing the FPGA chip, the vast difference in routing density in every RO would affect the performance. This work aims to find the ideal routing density of ROs to improve uniqueness. For this purpose, five logic arrangements in forming RO are tested to extract frequency differences over various routing densities. Then, the scenarios that narrow range routing density are designed, and ROs with a close value of routing density will be considered. Additionally, this work estimates the minimum runtimes of RO to avoid error and bit-flip.

## 3. Proposed RO-PUF

Research in finding and improving the quality of RO-PUF has long been carried out. In the conventional RO-PUF, the pulses generated by the ROs of a particular duration are detected by two counters simultaneously. The structure is equipped with a CRP generation process, where demultiplexers and multiplexers are placed before and after ROs, respectively. Therefore, the unequal route distances from ROs to counters caused inaccuracy in counting RO pulses, which were identified as locking phenomena [26,27] or jitter noise [28]. Meanwhile, this work proposes a modified RO-PUF structure that connects ROs directly to counters (no intermediary circuit between the ROs and the counters), as shown in Figure 1. The circuit of the CRP generation is placed next to the counters. This work has advantages, including:

- Routing equality—The distance between each RO to the respective counter is equal. Therefore, there will be no locking phenomena or jitter noise.
- One-time run—Each RO will only be run once. This technique would avoid the frequency difference of multiple RO run. In most conventional designs, every RO may be run more than once, leading to a slight difference in frequency. Moreover, multiple RO runs may result in less security due to a side-channel attack [29].
- Flexibility in choosing RO pairs—All CRP generation techniques proposed previously may be applied since the CRP process is independent, as shown in Figure 1.

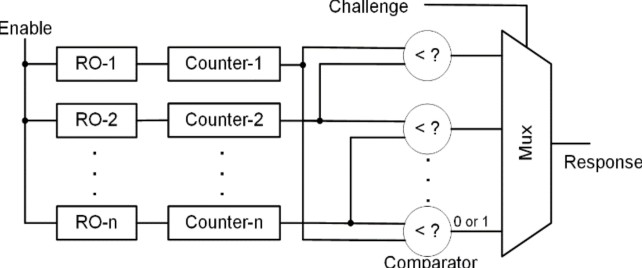

**Figure 1.** Propose RO-PUF using direct pulses count.

Putting a small number of ROs onto FPGA is a must to meet the practical condition of PUF as a hardware-based fingerprint. Therefore, this work realized $n = 30$ ROs to allow more space for the primary circuit. However, the proposed RO-PUF can generate more response bit ($n!/2(n-2)!$) compared to the conventional RO-PUF design by Suh et al. of $n/8$ response bit [20]. Researchers developed a new way of expanding response bit upon the CRP improvement while keeping small area usage. Maiti et al. [30] and Delavar et al. [31] are among the researchers who proposed expanding response bit techniques

where, for $n$ number of ROs, they can generate as many as $(2^n - n - 1)$ [30] and $(2^n - 1)$ [31] response bits. The CRP process is done outside the FPGA chip. Since the CRP generation of this proposed RO-PUF is independent, it may also be realized outside the FPGA chip.

### 3.1. RO Implementations

According to previous works, three-stage RO is less stable [28,32–34]. The frequencies of ROs are varied, from 120 MHz to 200 MHz, leading to a standard deviation of 11.3 MHz, which is relatively high compared to the ROs, which are realized using higher stages, around 1 MHz [32,34]. The resulting waveforms are challenging to detect by the counter [33]. Another work reported that the three-stage RO resulted in the worst reliability of 82.5% [28]. When RO is realized using more stages, the delay increases; consequently, the identification time would be longer, weakening its security. Therefore, this work chooses the five-stage RO.

Intel FPGA, previously Altera, is configured mainly using Quartus software. A particular modification is required to realize the ring oscillator onto the Cyclone V chip to prevent the circuit's streamline by Quartus during synthesis [26,28,35,36]. NAND gates were used to provide an alternative run of the desired period. A direct implementation of the VHDL code circuit was impossible because Quartus would streamline identical logics during synthesis. Hence, the circuit programmed using Quartus comprised of a NAND gate and LCELLs.

### 3.2. RO Placement

Maiti and Schaumont were the first to explore the RO placement location in the FPGA to see the frequency difference [21]. They investigated the RO performance of the Xilinx chip. Later, Feiten et al. explored several positions onto the Altera chip; they found that the RO at the corner of the chip had lower reliability [28]. Most of the previous attempts placed the amount of ROs in a location arranged in a 2-dimensional array. However, this study explores the variation of routing instead of RO position. For this purpose, ROs placed at different locations spread onto the entire chip. In this study, thirty ROs were located throughout the chip area. For similarity in designation, RO at locations 1, 2, . . . , 30 is written as RO-1, RO-2, . . . , RO-30. The counter was placed adjacent to the RO by utilizing a direct-link interconnect to guarantee route equality of all ROs to counters. This work, which lets Quartus located the remaining circuits function automatically, aims to have significant routing diversity among ROs.

## 4. Experimental Condition

This work set certain experiment conditions to extract response bits of the proposed RO-PUF. Those settings include logic arrangement and its LUT configuration of how RO is created inside the chip. Table 1 shows the experimental conditions used in conducting this research. This study extracts the statistical parameters and PUF quality within a limited time frame of up to 1 ms.

**Table 1.** Experimental conditions of the proposed RO-PUF.

| FPGA Technology | Intel Cyclone V (28 nm) |
| --- | --- |
| Chip No. | 5CSEMA5F31C6 and 5CSEMA4U23C6 |
| Software | Quartus Prime 17.1 |
| RO stage | Five-stage |
| Logic arrangement | Five patterns |
| RO runtimes | 100 ns–1 ms |
| Counter size | 20-bit |

### 4.1. Logic Arrangements

This work is the first to implement various logic arrangements in forming RO inside the FPGA chip. Different arrangements of logic may influence RO performance due to

routing density. Five different arrangements of logic (called patterns) are created. The LAB of Cyclone V consists of ten Adaptive Logic Modules (ALMs). There are two LUTs and several other components in an ALM. The gates forming RO will be placed on the LUTs. Figure 2 shows five variations in the position of the gate arrangement.

Figure 2a shows an arrangement according to Pattern-1. The gates were placed on the ALM-1, ALM-2, and ALM-3 in sequence or top to bottom. Meanwhile, Pattern-2 (as shown in Figure 2b) was the reverse sequence of Pattern-1. Pattern-3 was arranged using gates located in the middle of the LAB (ALM-5, ALM-6, and ALM-7), as shown in Figure 2c. Meanwhile, Pattern-4 (as shown in Figure 2d) was the opposite sequence of Pattern-3. The position of the gate placement following Pattern-5 was at the bottom of the LAB. Gates were placed on node ALM-10, ALM-9, and ALM-8, as shown in Figure 2e.

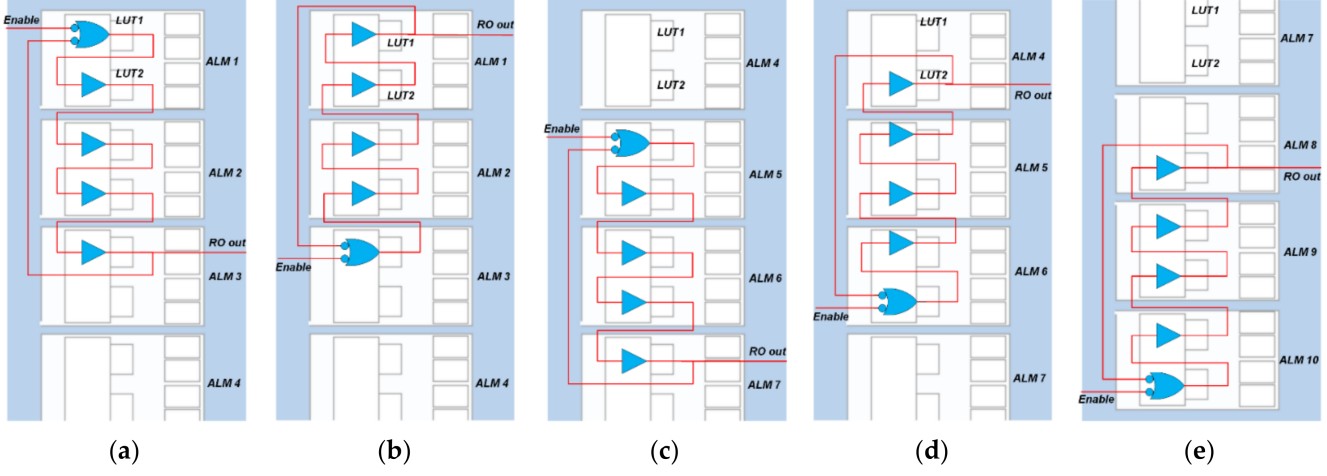

(**a**)  (**b**)  (**c**)  (**d**)  (**e**)

**Figure 2.** RO arrangements in a LAB (**a**) Pattern-1 (**b**) Pattern-2 (**c**) Pattern-3 (**d**) Pattern-4 (**e**) Pattern-5.

The study analyzed the configuration of all gates input forming RO in the LAB. Table 2 lists all input configurations of LUTs versus patterns assigned by Quartus. For instance, the input gates configuration of Pattern-2 is [DATAF DATAE-DATAF-DATAF-DATAF-DATAC] or FE-F-F-F-C, as shown in Table 2, of gates NAND-buffer-buffer-buffer-buffer.

**Table 2.** Configuration of LUT gate inputs in the Cyclone V.

| Arrangement | Input Configurations | RO Trigger |
|---|---|---|
| Pattern-1 | FB-D-C-F-F, DF-D-C-F-F, FB-A-C-F-F | F, D |
| Pattern-2 | FE-F-F-F-C | F |
| Pattern-3 | FC-F-F-F-F, FC-F-D-F-F, FC-F-C-F-F, DC-F-F-F-F, FD-F-F-F-F | F, D |
| Pattern-4 | FD-F-D-F-F, FA-F-D-F-F, DA-F-A-F-F | F, D |
| Pattern-5 | FD-F-D-F-F, FC-F-D-F-F, FD-B-A-F-F, FD-D-D-F-F | F |

All ROs arranged according to Pattern-2 are triggered via DATAF (F). More than one configuration is assigned to ROs of Pattern-1, Pattern-3, Pattern-4, and Pattern-5, as listed in Table 2. The assignments are due to the heavy routing around some ROs. Different input configurations of LUTs might impact the total delay of the RO. The delay is minimal when a buffer or NAND is configured via input DATAF or DATAE since the signal will only pass a multiplexer. Input DATAD resulted in a higher delay because the signal will pass two multiplexers or a four-input LUT (F0) and a multiplexer, while input DATAA, DATAB, or DATAC results in the highest delay.

### 4.2. Routing Density

In FPGA, connecting two logics separated into several blocks (LABs) may involve wires and switching circuits. Therefore, delays due to the interconnects are more significant than the circuit's logic delays [37]. Quartus assign efficient routing, which allows a design to be realized for minimizing space and delay. However, the problem of routing is architecture-dependent in different FPGA technologies available in the market. The row and column interconnect in the Intel FPGA vary on the technology. For Cyclone V, as shown in Figure 3a, the row routing uses R3, R6, and R14 interconnects, connecting logics traversing up to 3, 6, and 14 horizontal LABs, respectively [38]. Similarly, column routing is realized using C2, C4, and C12 interconnects. These interconnects (C2, C4, and C12) can connect logics traversing up to 2, 4, and 12 vertical LABs. Those rows and columns may be combined to create a route traversing few LABs of the row to column. In every interconnect type, half of them may be utilized to create a route to one direction only.

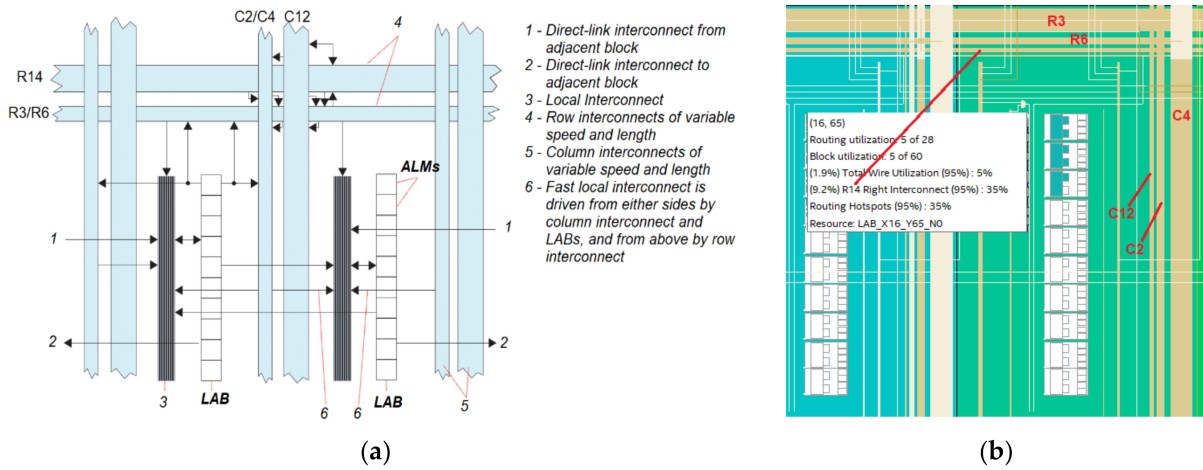

(a)   (b)

**Figure 3.** Overview of (**a**) Interconnects in Cyclone V (**b**) Routing density of RO-9 in Chip Planner (available in Quartus).

Besides row and column interconnect, a direct-link interconnect exists [38] in Cyclone V, which is dedicated to creating a route of logics of adjacent LAB. This work utilizes "wire utilization" and "routing hotspots" to estimate Intel FPGA chips' routing density [38,39]. The former refers to the percentage of wire utilized around a specific LAB, while the latter refers to the maximum percentage of any row or column interconnect of any direction. Routing hotspots are estimated based on the maximum total wire utilized of any direction of C2, C4, C12, R3, R6, or R14. Figure 3b views routing density of RO at location 9 (RO-9). The total wire used in this LAB is 5% (wire utilization). There are 28 wires available of R14 interconnect; 14 are to the right, and another 14 are to the left. Hence, the routing hotspots of this LAB are about 36% (5 of 14 wires).

This work prelocated the ROs and their counters. At the same time, other supporting circuits are not prelocated. As a result, routing around ROs differs significantly. Consequently, the wire utilization and routing hotspots of every ROs are also varied subjects to their location, where the ones in the middle of the chip are higher than others. Figure 4 shows the relationship between average wire utilization and routing hotspots. In general, wire utilization and routing hotspots are strongly related. Most of the higher wires utilized ROs, the routing hotspots were also higher, but this is not always true. For instance, in RO-15, when realized using Pattern-1, 6% of wires are used (wire utilization). The peak wiring (routing hotspots) in this RO is C12 down, where 10 out of 12 wires were utilized, which is equivalent to 83%. Conversely, in RO-14, 25% of wires are used, which is higher than RO-15. However, the routing hotspots are less than RO-15, 71% (10 out of 14 wires of the R14 left interconnect). These phenomena only occur in ROs of more than 5% wire utilization.

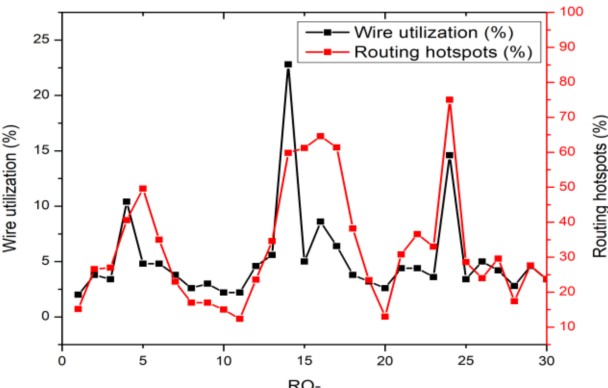

**Figure 4.** The relation between wire utilization and routing hotspots.

### 4.3. Metric Improvement

This study found that routing has a great influence on the frequency of the ROs; the higher routing around the RO location, the less its frequency is. The highest routing density is at RO-14 using all patterns. This RO's average wire utilization is 22.8% (the highest), and the average routing hotspots is 59.8%. Upon realization, the RO produced the least pulses in all chips' overall patterns, which leads to the worst uniqueness. This finding explained the lower uniqueness of some reported RO-PUF qualities [21,28,40]. The study found that other logic existed inside the LAB of RO-14, leading to heavier routings. Hence, the RO exposed noises more than others [23,24,41].

The unstable ROs were excluded to improve the PUF metric [5,42,43]. While in other designs, researchers considered only the RO pairs where their frequencies were far enough apart [43,44], Meaning that the RO inclusion process was based on the resulting frequencies upon realization, varying from chip to chip, this work, on the other hand, improves the PUF metric (uniqueness) by excluding ROs based on routing density. The ROs are chosen before they are realized onto the chip. The ROs with heavy and light routing densities are excluded, or only ROs with narrow routing densities will be considered. For this purpose, this work applies four scenarios in excluding ROs.

- The first scenario (named WR1); excludes ROs with heavy wire utilization ($w_h$).
- The second scenario (named WR2); excludes ROs with heavy wire utilization ($w_h$) and light wire utilization ($w_l$).
- The third scenario (named RH1); exclude ROs with heavy routing hotspots ($r_h$).
- The fourth scenario (named RH2); exclude ROs with heavy routing hotspots ($r_h$) and light routing hotspots ($r_l$).

The thresholds ($w_h$, $w_l$, $r_h$, $r_l$) for particular ROs are subject to RO-PUF design. Under these scenarios, fewer ROs are needed to determine the metrics. The thresholds in this work are applied equally to all patterns. This work chooses the thresholds of $w_h$ = 5%, $w_l$ = 3%, $r_h$ = 50%, $r_l$ = 20%, based on the medians of wire utilization (4%) and routing hotspots (28.1%).

## 5. Results and Discussion

The proposed RO-PUF is verified using ten Intel Cyclone V chips (5CSEMA5F31C6 and 5CSEMA4U23C6) in the DE1-SoC and DE0-Nano board. Quartus is used to create and verify the design before downloading it onto the FPGA chip. The performance is analyzed over various routing densities of ROs and patterns. The study further evaluates the shift of performance by applying scenarios. The proposed RO-PUF can be applied to other FPGA chips or platforms with few modifications. The use of direct-link interconnects to guarantee routing equality may be replaced by direct connections or direct lines in Xilinx chips [45]. Another adjustment is that the counter size should be fitted to the maximum number of available flip-flops inside a LAB or configurable logic block (CLB) or to use a ripple counter when a counter is realized with more than one CLB.

## 5.1. Statistical Properties

This study evaluates the average, range, deviation, and standard deviation of frequency upon pulses resulted among patterns. The average frequencies in a higher temperature are lower, indicating that ROs generated fewer pulses than those at low temperatures [3]. When ROs are realized using Pattern-3 at low temperatures, they produce the highest average frequency (428.28 MHz) compared to other patterns. Meanwhile, the ROs implemented using Pattern-1 resulted in the least average frequency (354.64 MHz). The vast difference has occurred because Quartus assigned different LUT inputs that form RO among patterns, as shown in Table 2. Another reason is due to the different positioning of input and output connections of LAB of different patterns, which indirectly determine average routing hotspots among the patterns. For instance, the average routing hotspots of Pattern-1, at 35.23%, is higher than Pattern-3, 29.77%. Hence, the delay of ROs concerning the pattern is varied, which leads to the difference in average frequencies.

The range of frequencies of the ROs resulted among patterns also vary. The most extensive range resulted using Pattern-4, which is 130.05 MHz, and the narrowest range resulted using Pattern-2, which is 37.36 MHz. The narrow frequency range of Pattern-2 is due to all ROs being assigned the same input configuration, as listed in Table 2. The study estimated the frequency deviation and maximum frequency deviation of the pulse to understand RO behavior better. The frequency deviations of Pattern-1, Pattern-2, Pattern-3, Pattern-4, and Pattern-5 are 4.45 MHz, 1.29 MHz, 2.80 MHz, 4.48 MHz, and 4.15 MHz, respectively. In comparison, the maximum deviations of Pattern-1, Pattern-2, Pattern-3, Pattern-4, and Pattern-5 are 37.69 MHz, 6.78 MHz, 22.13 MHz, 58.22 MHz, and 44.95 MHz, respectively. The vast differences between maximum deviation and average deviation among patterns indicate that the number of pulses generated by the RO is due to delay and the difference of routing density.

The study found a slight increase (1.62%) of the average frequency among patterns when scenarios are applied. Meanwhile, the frequency ranges and standard deviations decreased in most scenarios, as shown in Figure 5. There is a significant reduction of frequency range except using Pattern-2 (all scenarios) and Pattern-3 (scenario RH1 and RH2), as shown in Figure 5a. The significant change of standard deviation (31.2%) between scenarios occurred when the RO was realized using Pattern-3, Pattern-4, and Pattern-5, as shown in Figure 5b. These changes might affect the increase of uniqueness and the reduction in reliability. The list of frequency ranges and standard deviations and their change can be seen in Table 3.

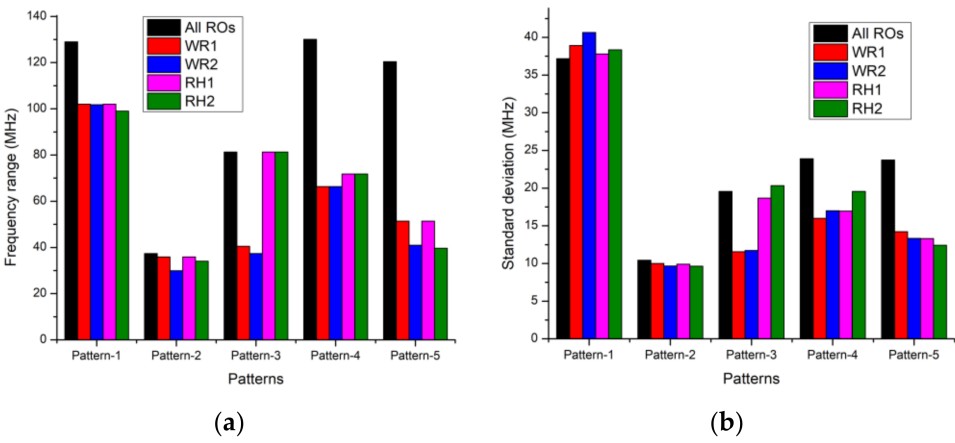

(a) (b)

**Figure 5.** The shift upon four scenarios of (**a**) frequency range (**b**) standard deviation.

**Table 3.** Statistical properties and RO-PUF metrics of logic arrangements and scenarios.

|  | Scenarios | Pattern-1 | Pattern-2 | Pattern-3 | Pattern-4 | Pattern-5 |
|---|---|---|---|---|---|---|
| Freq. range (MHz) | All ROs | 128.98 | 37.36 | 81.33 | 130.05 | 120.43 |
| (Smaller is better) | WR1 | 102.01 | 35.95 | 40.48 | 66.38 | 51.38 |
|  | WR2 | 101.75 | 29.95 | 37.36 | 66.38 | 40.99 |
|  | RH1 | 102.01 | 35.95 | 81.33 | 71.83 | 51.38 |
|  | RH2 | 99.06 | 34.07 | 81.33 | 71.83 | 39.70 |
| Standard Deviation (MHz) | All ROs | 37.18 | 10.42 | 19.57 | 23.90 | 23.73 |
|  | WR1 | 38.92 | 10.00 | 11.55 | 15.98 | 14.22 |
|  | WR2 | 40.64 | 9.66 | 11.72 | 16.99 | 13.32 |
|  | RH1 | 37.80 | 9.91 | 18.67 | 16.95 | 13.31 |
|  | RH2 | 38.34 | 9.64 | 20.32 | 19.56 | 12.43 |
| Uniqueness (%) | All ROs | 34.58 | 52.71 | 46.90 | 44.32 | 41.31 |
|  | WR1 | 30.77 | 53.87 | 52.68 | 46.67 | 49.11 |
|  | WR2 | 37.66 | 54.58 | 53.24 | 48.87 | 49.60 |
|  | RH1 | 35.96 | 53.32 | 48.40 | 48.62 | 38.85 |
|  | RH2 | 41.12 | 52.51 | 50.18 | 37.58 | 42.34 |
| Reliability (%) | All ROs | 99.47 | 98.93 | 99.15 | 99.04 | 99.12 |
|  | WR1 | 99.37 | 98.64 | 98.95 | 98.88 | 99.18 |
|  | WR2 | 99.51 | 98.21 | 98.76 | 99.18 | 98.85 |
|  | RH1 | 99.28 | 98.74 | 98.61 | 98.99 | 98.63 |
|  | RH2 | 99.20 | 98.56 | 98.34 | 98.75 | 98.58 |
| Uniformity (%) | All ROs | 38.82 | 46.22 | 47.55 | 43.81 | 58.59 |
|  | WR1 | 34.05 | 45.91 | 46.65 | 33.39 | 54.66 |
|  | WR2 | 33.78 | 43.04 | 41.85 | 41.32 | 47.02 |
|  | RH1 | 34.86 | 47.13 | 45.77 | 44.48 | 58.19 |
|  | RH2 | 33.12 | 42.08 | 43.64 | 33.93 | 62.98 |

### 5.2. RO-PUF Metric

This study examines the RO-PUF metric in terms of uniqueness, reliability, and uniformity. The uniqueness is calculated according to the difference in responses bits among FPGA boards (chips) using hamming distance (*HD*) calculated by Equation (1) [46], where $c$ is the number of the chip, $r_b$ is the number of response bits, $R_u$ and $R_v$ are the response bits of $u$ and $v$ boards being compared, $HD(R_u, R_v)$ is a hamming distance between the response bits.

$$Uniqueness = \frac{2}{c(c-1)} \sum_{u=1}^{c-1} \sum_{v=u+1}^{c} \frac{HD(R_u, R_v)}{rb} \times 100\%, \tag{1}$$

Digital circuit performance is influenced by environmental factors such as temperature, supply voltage, noise, etc. In this study, the performance of RO-PUF subsequent to temperature changes will be observed. The frequency of RO decreases when the temperature is increased [20,47,48]. Therefore, this work calculated reliability based on the consistency of pulses at different temperatures according to Equations (2) and (3), where $k$ is the number of experiments on the same chip, $R_s$ is the response bit from chip $i$ at low temperature, $R_{s,t}$ is the $t$-th sample of $R'_s$ response bit from chip $i$ at a higher temperature.

$$HD\ Intra = \frac{1}{k} \sum_{t=1}^{k} \frac{HD(R_s, R'_{s,t})}{rb} \times 100\%, \tag{2}$$

$$Reliability = 100\% - HD\ Intra, \tag{3}$$

Uniformity is used to measure the ratio between "1" and "0" in a series of bit responses that determine the randomness of a specific chip's responses. The ideal uniformity is 50%, which means the number of "1" and "0" in a series of response bits is equal. This work calculated the uniformity by comparing the hamming weight (*HW*) of "1" on the same

chip, according to Equation (4), where $R_{s,l}$ is the $l$ bit from the sequence of response bits of a particular chip.

$$Uniformity = \frac{1}{rb} \sum_{l=1}^{rb} R_{s,l} \times 100\%, \tag{4}$$

This work generates response bits by providing all possible pairs' challenges where the maximum number of pairs is determined by a combination of 2 among $n = 30$ ROs, which results in $r_b = 435$ bits response. The differences in responses at low (25 °C) and high (50 °C) temperatures are used to determine reliability. This work avoids temperatures higher than 50 °C to prevent the chip's performance degradation since the FPGA board is tested repeatedly over various patterns using various runtimes.

This study found that uniqueness and reliability are varied among patterns, as listed in Table 3. The realization of using Pattern-2 resulted in the best uniqueness of 52.71%. It agrees with the narrow frequency range and low standard deviation of Pattern-2, as shown in Figure 5. While the RO arrangement, according to Pattern-1, produces the lowest uniqueness. Regarding reliability, Pattern-1 resulted in the highest score, while Pattern-2 produced the lowest score. However, the difference in reliability of all patterns is less than 1%.

The uniqueness and reliability are subject to change upon scenarios, as shown in Figure 6. The uniqueness of all patterns and their shift upon scenarios are shown in Figure 6a. While Figure 6b viewed the reliability of all patterns and their shift when the scenarios are applied. In most scenarios, the uniqueness increase, and the reliability decrease compared to when all ROs are involved. The study identified unusual issues among scenarios where the uniqueness decreases and the reliability increase. The first issue, uniqueness, is decreased when ROs implemented using Pattern-1, Pattern-4, and Pattern-5 using WR1, RH2, and RH1 scenarios, respectively. The second issue was the increased reliability when the ROs realized using Pattern-1, Pattern-4, and Pattern-5 using WR2, WR2, and WR1. The issues occurred because the presented data in Figure 6 show the average of all runtimes; second, some excluded ROs may be unstable [5,42,43].

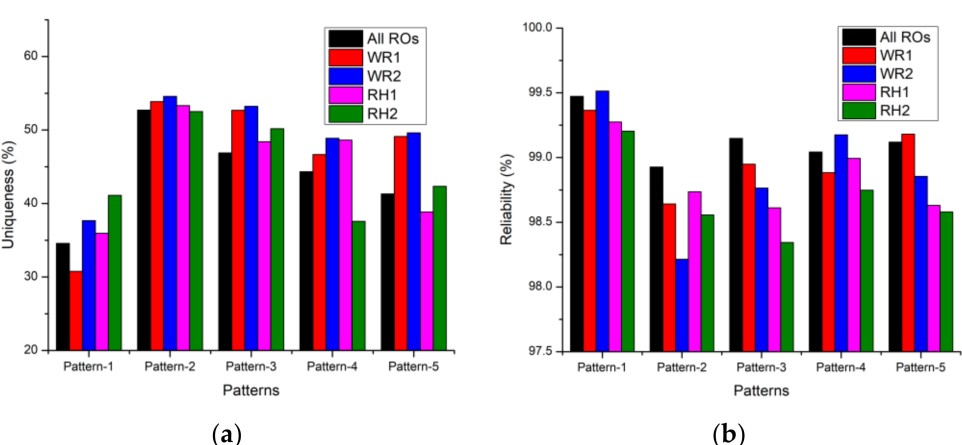

**Figure 6.** The shifts upon four scenarios of (**a**) uniqueness (**b**) reliability.

Unlike uniqueness and reliability, the uniformity trend is generally varied over patterns and scenarios, as shown in Figure 7. The uniformity of Pattern-2 and Pattern-3 is close to the ideal value of 50%. The realization using Pattern-1 resulted in the least random. The uniformity of RO-PUF when Pattern-4 and Pattern-5 are applied is better than Pattern-1 but not as random as Pattern-2 and Pattern-3. Most of the patterns produced more "1" in a series of responses, but Pattern-5 resulted in more "0". Upon the scenario implementations, most of them resulted in less random outcomes. Table 3 shows the performance and statistical properties for different patterns. Varying performance among patterns is due to different LUT input configurations, which in turn affect the delay of RO among patterns.

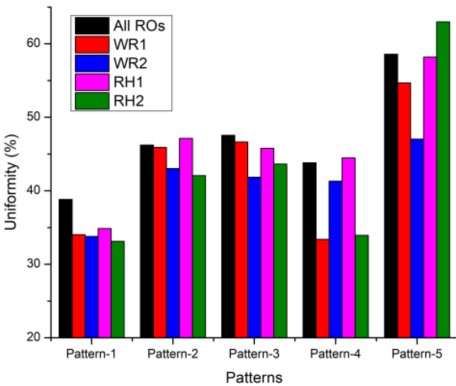

**Figure 7.** Uniformity and its shifts over scenarios.

*5.3. Result Comparisons*

Table 4 lists the area utilization for creating RO and metric comparison of RO-PUFs. The previous works improved the PUF quality by using different techniques [12,28,30,49], such as applying gray code to the counter value [12]. This study compares LUT utilization for implementing ROs, where this work requires 150 LUTs, which is the least compared to other works with higher LUT utilization. As a comparison for $r_b$ = 28, and by assuming RO or counter fits onto one LAB, the proposed structure would require 8 LABs for ROs + 8 LABs for counters + 28 of 2-to-1 MUXs. Meanwhile, in the conventional design [29], in contrast, it would require 56 LABs for ROs + 2 LABs for counters + 2 of 28-to-1 MUXs). Then, by assuming the resource for MUXs is equal, the conventional structure requires 58 LABs, while the proposed method is about 16 LABs. Furthermore, the CRP of the proposed structure is independent, which means it could be realized outside the FPGA chip to enhance response bits. From this work, additional gates are required for realizing other circuits. Upon implementation of RO-PUF onto Cyclone V, this work requires 702 ALMs plus 794 registers. However, most of the utilized gates are due to control circuits. Upon realization of a fixed runtime, area utilization may be reduced. An additional area is required for CRP generation; however, the total area occupied is less than 5% (DE1-SoC) and 7% (DE0-Nano). The design would draw approximately 420 mW of static power, as estimated using the Power Analyzer Tool (available in Quartus).

**Table 4.** Area utilization for creating RO and metrics comparison of RO-PUFs.

| | ROs (LUTs) | Uniqueness (%) | Reliability (%) | Uniformity (%) | Platform |
|---|---|---|---|---|---|
| Suh et al. [20], 2007 | 1024 (6144) | 46.15 | 99.52 | - | Xilinx |
| Maiti et al. [21], 2009 | 256 (1280) | 35.91–45.90 | - | - | Xilinx |
| Merli et al. [49], 2010 | 129 (-) | 43.40−48.51 | 99.20, 98.28 | - | Xilinx |
| Xin et al. [40], 2011 | 64 (3008) | 32, 41 | 99.29 | - | Xilinx |
| Maiti et al. [30], 2011 | 16 (256) | 49.99−50.07 | ±92 *, ±70 * | 50.02, 49.4 | Xilinx |
| Feiten et al. [28], 2013 | 64 (1024) | 6.68−37.03 | 99.41–82.5 | 50.00, 62.07 | Altera |
| Sahoo et al. [50], 2013 | 75 (1200) | 47.57 | 90.70 ** | 47 | Altera |
| Kodytek et al. [12], 2016 | 900 (6300) | 48.42−48.74 | 98.22, 97.55 | - | Xilinx |
| Delavar et al. [31], 2016 | 512 (775) | 49.81 | 96.07 | - | Xilinx |
| Chauhan et al. [51], 2019 | 11,264 (-) | 49.9 | 97.85–99.80 | - | Xilinx |
| Deng et al. [52], 2020 | 128 (2048) | 49.95 | 91.4–99.13 * | 49.61 | Xilinx |
| This work | 30 (150) | 50.18 # | 99.51 ## | 47.55 | Intel (Altera) |

\* Approximation, \*\* Reliability measured in room temperature, # Pattern-3, ## Pattern-1 (WR2).

Comparing RO-PUFs, this study did not classify or identify the impact of aging, which may shift the frequency [53–55]. For instance, a high-temperature setup and its duration for extracting data from chips may degrade the frequency—this work extracts a frequency at a temperature no higher than 50 °C. In general, the result of this work is better compared to the previously proposed methods using a similar platform (Altera), and the performance

is as good as the latest works implemented onto Xilinx [51,52]. Upon a similar FPGA platform (Altera/Intel), this work is the first to realized RO-PUF onto Cyclone V (28 nm). In 2012, Bernard et al. implemented the ROs using Cyclone II (90 nm) and Cyclone III (65 nm) [26], but there is no performance metric reported. Later, Sahoo et al. utilized their design onto Cyclone III (65 nm) [50], where the reported uniqueness is slightly closer to the ideal (47.57%) but has poor reliability (90.70%). Later, Feiten et al. implemented RO-PUF onto Cyclone IV (60 nm) [28]. However, the resulting uniqueness of 37.03% and the reliability of 82.5% is relatively smaller than that found in this work, as listed in Table 4.

*5.4. Runtimes Consideration*

Most of the RO-PUF designs in the literature did not specify the RO runtime in detail. Mustafa et al. mentioned that the RO runtime is 0.14 ms [32], and Feiten et al. enabled the series around 20 ms [28]. The analysis of the runtime was carried out by Merli et al. [49]. The ROs were run in the range of 200 ns to 204.8 μs. Therefore, this work expands individual RO ($t_{RO}$) runtimes with a duration varying from 100 ns to 1 ms. It is estimated that up to 1 ms, the number of pulses generated by RO has not exceeded the counter capacity. The ROs were run one by one with a pause of $d_i$ = 100 ns. This technique was chosen to avoid interference between ROs or other adjacent circuits. Therefore, the total time ($t_{total}$) required to activate *n* ROs is formulated according to Equation (5).

$$t_{total} = (n\, t_{RO}) + (n-1)d_i \,, \tag{5}$$

In the previous study by Merli et al. [49], an error would occur if the RO were run in a short duration. The author recommended using at least 12.8 μs. Therefore, this study investigates this issue to determine the minimum runtime because of using different FPGA technology. In short duration runtimes, the study found that ROs generated pulses in close range. For example, upon realization using Pattern-1 at chip-1 at low temperature, for a runtime of 100 ns, ROs generated ranging from 30 to 45 pulses. The pulses generated by ROs were mostly equal. Then, for a runtime of 200 ns, the produced pulses were slightly different, ranging from 60 to 90. Upon further longer runtimes of 2 μs, the number of pulses produced varied among ROs in close proximities. Some RO pairs resulted in equal pulses, such as RO-4 and RO-7, RO-5 and RO-9, RO-2 and RO-10, RO-27, and RO-29, which generated 690, 691, 696, 735 pulses, respectively.

Whenever two or more ROs generate equal pulses, this leads to bit-flip when the ROs are run on different environmental conditions. For the use as a PUF, all ROs should produce different pulses that are scattered away from each other. Therefore, this work suggests using the minimum runtimes ($t_{RO\_min}$) of individual RO to guarantee at least *s* pulses difference among ROs using Equation (6).

$$t_{RO\_min} = s/\Delta f_{min}, \text{ where } s = 1, 2, 3, \tag{6}$$

where $\Delta f_{min}$ refers to the minimum frequency difference among ROs based on the experiment, the minimum frequency of the recorded data in this work is 0.105 MHz, which means that the minimum runtimes guarantee at least *s* = 1 pulse difference is around 9.52 μs. In the previous works [43,44], a threshold of $\Delta f_{min}$ was set to improve reliability. Any RO pairs with a frequency difference less than the set threshold are ignored, reducing the number of response bits. However, instead of setting a threshold of frequency difference of RO pairs, this work suggests increasing the runtimes until ROs produce minimum *s* pulses difference. Hence, all RO pairs may be used, and a maximum response bit may be generated.

## 6. Conclusions

Implementation and analysis of the area-efficient RO-PUF based on direct pulses count were successfully carried out. The proposed RO-PUF requires a lesser area compared to other works to realized ROs. The total area required for realizing the proposed RO-PUF onto DE1-SoC (5CSEMA5F31C6) and DE0-Nano (5CSEMA4U23C6) is less than 5% and 7%,

respectively. The PUF metrics and statistical properties are calculated over routing diversity using five RO arrangements. Over scenarios that limit the range of routing density (wire utilization and routing hotspots) closer to the median, the uniqueness increased about 2%. The best uniqueness and reliability were 50.18% and 99.51%, respectively. Therefore, the proposed structure's PUF metrics (uniqueness and reliability) are much better than the works realized on Altera, and it is as good as the recent RO-PUF realized on Xilinx. Moreover, this work suggests the minimum runtimes of RO using Equation (6) to reduce error and response bit-flip.

**Author Contributions:** Methodology, Z.Z., N.S., S.F.W.M.H.; software, Z.Z.; formal analysis, Z.Z., N.S., S.F.W.M.H.; investigation, Z.Z., N.S., S.F.W.M.H., M.S.A.T.; resources, Z.Z., S.F.W.M.H., A.J.; writing—original draft preparation, Z.Z.; writing—review and editing, Z.Z., N.S., S.F.W.M.H., M.S.A.T., A.J.; visualization, Z.Z., S.F.W.M.H. All authors have read and agreed to the published version of the manuscript.

**Funding:** This research received no external funding.

**Institutional Review Board Statement:** Not applicable.

**Informed Consent Statement:** Not applicable.

**Data Availability Statement:** Not applicable.

**Conflicts of Interest:** The authors declare no conflict of interest.

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
