# Peer review of "Routing Density Analysis of Area-Efficient Ring Oscillator Physically Unclonable Functions"

_applsci, doi:10.3390/app11209730_

Round 1

Reviewer 1 Report

The authors focus on the implementation method of RO_PUF in FPGA (Quartus) structures. One of the goals of an effective implementation is to limit the use of logical resources (ALM blocks). The article is interesting and contains a significant number of experiments for individual implementations.

The main problem that makes it difficult to fully understand the article is chapter 4.2, and more precisely Figure 2, which does not fully reflect the description in this chapter. I encourage the authors to correct Figure 2 and the content in chapter 4.2 in such a way that all the symbolic symbols introduced, abbreviations in the content of the chapter, are reflected in the drawing. 

Reviewer 2 Report

The paper introduces a new design of RO-PUF, which connects ROs directly to the counters. Moreover, it presents a detailed step-by-step analysis of basic PUF metrics regarding various arrangements in FPGAs and highly impactful routing density. The study is described clearly and carefully, with high technical detail.

Despite the positive general reception, I would recommend the authors to respond to the following questions:

  1. In Table 3 and the following description, the authors focused on LUT utilization in the RO implementation. However, the solution proposed in section 3 significantly increases the number of counters necessary to support the ROs, which translates directly into greater hardware usage. This is an important aspect in low-complexity devices with limited resources on a dedicated security structure. How does the hardware cost of the reduction in connection length compare to the total hardware requirements for a conventional RO-PUF with an ROs-MUX-counter structure?
  2. Subsection 3.1 lacks basic information, forcing to jump over references, i.e., describing three-stage RO as less stable requires a simple indication of some reference structure. Likewise, frequency deviation requires the indication of an average value against which it becomes significant.
  3. Subsection 4.1 describes different RO arrangements in LABs justified by differences in resulting routing density. It would be beneficial to present such a relation as it was not directly indicated in the results.
  4. Routing description (4.2) and second paragraph of results comparison (5.3) highly suggest a strong dependence of the obtained results on the tested platform, i.e., Cyclon V. Can the obtained results be generalized to avoid their applicability only to the viability of Cyclon V?
